# Performance of Three Tests for SARS-CoV-2 on a University Campus Estimated Jointly with Bayesian Latent Class Modeling

T. Alex Perkins,[a] Melissa Stephens,[b] Wendy Alvarez Barrios,[c] Sean Cavany,[a] Liz Rulli,[c] Michael E. Pfrender[a]

[a]Department of Biological Sciences, University of Notre Dame, Notre Dame, Indiana, USA
[b]Genomics and Bioinformatics Core Facility, University of Notre Dame, Notre Dame, Indiana, USA
[c]Notre Dame Research, University of Notre Dame, Notre Dame, Indiana, USA

**ABSTRACT** Accurate tests for severe acute respiratory syndrome coronavirus 2 (SARS-CoV-2) have been critical in efforts to control its spread. The accuracy of tests for SARS-CoV-2 has been assessed numerous times, usually in reference to a gold standard diagnosis. One major disadvantage of that approach is the possibility of error due to inaccuracy of the gold standard, which is especially problematic for evaluating testing in a real-world surveillance context. We used an alternative approach known as Bayesian latent class modeling (BLCM), which circumvents the need to designate a gold standard by simultaneously estimating the accuracy of multiple tests. We applied this technique to a collection of 1,716 tests of three types applied to 853 individuals on a university campus during a 1-week period in October 2020. We found that reverse transcriptase PCR (RT-PCR) testing of saliva samples performed at a campus facility had higher sensitivity (median, 92.3%; 95% credible interval [CrI], 73.2 to 99.6%) than RT-PCR testing of nasal samples performed at a commercial facility (median, 85.9%; 95% CrI, 54.7 to 99.4%). The reverse was true for specificity, although the specificity of saliva testing was still very high (median, 99.3%; 95% CrI, 98.3 to 99.9%). An antigen test was less sensitive and specific than both of the RT-PCR tests, although the sample sizes with this test were small and the statistical uncertainty was high. These results suggest that RT-PCR testing of saliva samples at a campus facility can be an effective basis for surveillance screening to prevent SARS-CoV-2 transmission in a university setting.

**IMPORTANCE** Testing for severe acute respiratory syndrome coronavirus 2 (SARS-CoV-2) has been vitally important during the COVID-19 pandemic. There are a variety of methods for testing for this virus, and it is important to understand their accuracy in choosing which one might be best suited for a given application. To estimate the accuracy of three different testing methods, we used a data set collected at a university that involved testing the same samples with multiple tests. Unlike most other estimates of test accuracy, we did not assume that one test was perfect but instead allowed for some degree of inaccuracy in all testing methods. We found that molecular tests performed on saliva samples at a university facility were similarly accurate as molecular tests performed on nasal samples at a commercial facility. An antigen test appeared somewhat less accurate than the molecular tests, but there was high uncertainty about that.

**KEYWORDS** Bayesian statistics, COVID-19, RT-PCR, SARS-CoV-2, epidemiology, molecular diagnostic, public health surveillance

**M**olecular testing has played a vital role in efforts to suppress the transmission of SARS-CoV-2. This applies in both community settings (1, 2) and in more specialized settings, such as hospitals (3, 4), workplaces (5, 6), schools (7, 8), and travel (9, 10). Although many contextual factors affect the success of testing (11, 12), the foundation

Address correspondence to T. Alex Perkins, taperkins@nd.edu, or Michael E. Pfrender, mpfrende@nd.edu.

The authors declare no conflict of interest.

of any successful testing program is the availability of tests that are sufficiently sensitive and specific to achieve the program's objectives.

Most evaluations of the sensitivity and specificity of molecular tests for SARS-CoV-2 have been performed in reference to a diagnostic that was considered a gold standard (13). Designating a diagnostic as a gold standard makes the calculation of sensitivity and specificity straightforward, as true-positive (TP), true-negative (TN), false-positive (FP), and false-negative (FN) test outcomes can all be defined clearly in reference to the gold standard. Under this assumption, the sensitivity can be estimated as TP/(TP + FN) and specificity as TN/(TN + FP).

A key limitation of this approach is that the estimates it yields are only as reliable as the gold standard on which they are based. The most common gold standard is reverse transcriptase PCR (RT-PCR) testing (14). This standard is far from golden, however. Especially with respect to sensitivity, the performance of these tests for SARS-CoV-2 has been found to vary as a function of the method of sample extraction (15, 16), day of infection (17, 18), and disease severity of the subject (19). Furthermore, designation of one method as a gold standard makes it impossible to evaluate whether another test might actually have better sensitivity or specificity than the presumed gold standard (20).

One way to circumvent the limitations associated with relying on a gold standard is to use an alternative method for analysis, such as Bayesian latent class modeling (BLCM) (21). This method involves joint estimation of the sensitivity and specificity of each type of test used, by virtue of considering the possibility that any given test result could have been erroneous for some, all, or none of the tests used. While this approach does not make test results more accurate *per se*, it does reduce the risk of bias associated with erroneously assuming that a gold standard is without error. This approach has been applied in some cases for molecular tests for SARS-CoV-2, resulting in differences relative to estimates that relied on a gold standard (22–24). For example, in a meta-analysis comparing RT-PCR testing of nasopharyngeal and saliva samples, allowing for imperfections in both types of tests resulted in higher estimates of specificity and narrower uncertainty about sensitivity (23).

In this study, we applied BLCM to a data set from a SARS-CoV-2 testing program in a university setting during October 2020. A unique feature of this data set is that it includes both RT-PCR and antigen tests, which have not been compared in previous BLCM analyses for SARS-CoV-2 that we are aware of (22–24). Another unique feature of this data set is that the majority of subjects were tested during surveillance screening and were not suspected of being infected at the time of testing. This presents an opportunity to quantify the test performance in a context that is highly relevant for public health (11). Moreover, the fact that the majority of subjects were in the 18-to-25 age range presents an opportunity to quantify test performance in a population for which tests could potentially be less sensitive (19, 25, 26) yet are of high value for surveillance (7, 8).

## RESULTS

Combining data across all tests performed, the test positivity was 2.5% (43/1,716). The positivity was lower among individuals tested for surveillance purposes (1.9%) than among individuals tested for other reasons (12.3%). The positivity was also lower for commercial tests (1.4%) than for saliva (3.1%) and antigen (13.5%) tests. The lower positivity among commercial tests held when controlling for the method by which individuals came to participate in the study (Table S1). Despite the differences in positivity, the very low positivity overall meant that the concordance was high: 99.3% between commercial and saliva tests, 96.3% between commercial and antigen tests, and 97.3% between saliva and antigen tests.

Using Bayesian analysis, we estimated a total of eight parameters (Table 1) by leveraging joint information about all observed combinations of testing outcomes across the three types of tests (Table 2). Application of this method to 100 simulated data sets showed good coverage of the true parameter values (Fig. S2 in the supplemental material).

**TABLE 1** Parameter definitions and posterior estimates

| Symbol | Definition | Posterior median (95% CrI)[a] |
|---|---|---|
| $Se_{Comm}$ | Sensitivity of commercial PCR test | 0.859 (0.547–0.994) |
| $Se_{Saliva}$ | Sensitivity of saliva test | 0.923 (0.732–0.996) |
| $Se_{Antigen}$ | Sensitivity of antigen test | 0.748 (0.373–0.969) |
| $Sp_{Comm}$ | Specificity of commercial test | 0.998 (0.992–0.999) |
| $Sp_{Saliva}$ | Specificity of saliva test | 0.993 (0.983–0.999) |
| $Sp_{Antigen}$ | Specificity of antigen test | 0.978 (0.888–0.999) |
| $Prev_{Non-surv}$ | Prevalence among nonsurveillance samples | 0.141 (0.058–0.258) |
| $Prev_{Surv}$ | Prevalence among surveillance samples | 0.018 (0.009–0.030) |

[a]Decimal values are provided, consistent with the definitions of these quantities as probabilities, rather than percentages, in Materials and Methods.

Applying the method to empirical data demonstrated good convergence (Fig. S3) and resulted in posterior samples with moderately low correlation (Fig. S4), suggesting that the data were reasonably informative about the parameters we sought to estimate.

The prevalence inferred by our Bayesian analysis was similar to the test positivity for surveillance testing (median, 1.8%; 95% credible interval [CrI], 0.9 to 3.0%) (Fig. 1A) and slightly higher for nonsurveillance testing (median, 14.1%; 95% CrI, 5.8 to 25.8%) (Fig. 1B). For the saliva tests, we estimated a sensitivity of 92.3% (95% CrI, 73.2 to 99.6%) (Fig. 1C, green) and a specificity of 99.3% (95% CrI, 98.3 to 99.9%) (Fig. 1D, green). Had we considered the commercial tests to be the gold standard, we would have instead estimated the sensitivity and specificity of the saliva tests to be 83.3% and 99.5%, respectively. Similarly, for the antigen tests, our estimate of the sensitivity (74.8%; 95% CrI, 37.3 to 96.9%) (Fig. 1C, blue) was greater than an estimate made in reference to the commercial tests (0.5), and our estimate of the specificity (97.8%; 95% CrI, 88.8 to 99.9%) (Fig. 1D, blue) was lower than an estimate made in reference to the commercial tests (100%). It is important to note the large uncertainty around these estimates due to the relatively low number of individuals who received an antigen test. These discrepancies were a result of the fact that we did not consider commercial tests to be the gold standard and estimated their sensitivity and specificity alongside those of the other two test types. Doing so resulted in estimates of the sensitivity of 85.9% (95% CrI, 54.7 to 99.4%) (Fig. 1C, red) and specificity of 99.8% (95% CrI, 99.2 to 99.9%) (Fig. 1D, red) for the commercial tests.

While comparison of the median sensitivities and specificities implies that some tests were more sensitive or specific than others, the wide uncertainty of our estimates must be considered when making such comparisons. We obtained more nuanced insight into the relative sensitivities and specificities of the three tests by calculating the proportion of samples in which the sensitivity of one test exceeded that of another,

**TABLE 2** Testing data[a]

| Reason for testing | Commercial | Saliva | Antigen | No. of participants |
|---|---|---|---|---|
| Surveillance | - | - | - | 3 |
| | - | - | NA | 778 |
| | - | NA | NA | 7 |
| | NA | - | NA | 3 |
| | - | + | NA | 4 |
| | + | - | NA | 1 |
| | + | + | NA | 9 |
| | NA | + | NA | 7 |
| Other | - | - | - | 22 |
| | - | - | NA | 7 |
| | NA | - | - | 5 |
| | NA | + | + | 4 |

[a]Each of the 853 study participants fell into one of the categories represented by each row. These categories differed with respect to the reason for testing and the outcome of each test. NA, not applicable; this indicates that a given test was not performed for those individuals. This table constitutes the full information used in our analysis. −, symbol indicates a negative test; +, indicates a positive test.

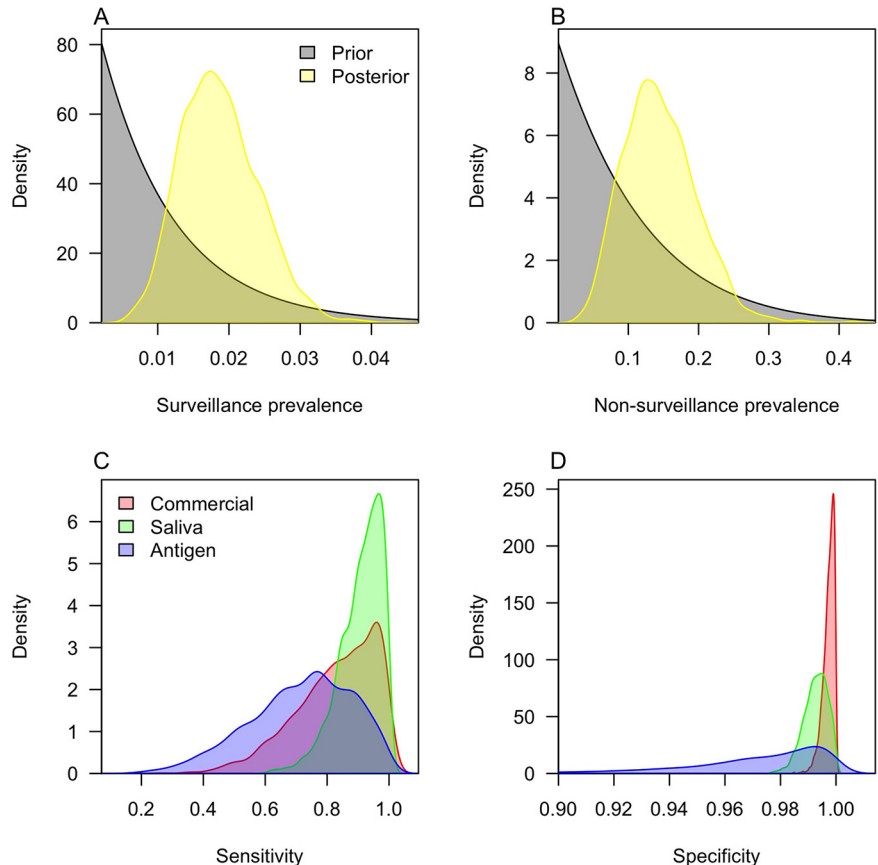

**FIG 1** Posterior parameter estimates. (A) Prevalence among individuals participating in surveillance testing; (B) prevalence among individuals participating in testing for reasons other than surveillance; (C) test sensitivity; and (D) test specificity. The colors in A and B distinguish the prior from posterior distributions, and the colors in C and D distinguish the different types of tests. Values outside the range 0 to 1 occur only as a result of smoothing. Decimal values are shown along the x axes, consistent with the definitions of these quantities as probabilities, rather than percentages, in Materials and Methods.

and likewise for the specificity. On that basis, we found a probability of 0.69 that the saliva test was more sensitive than the commercial test (Table S2). The probabilities that the saliva and commercial tests were more sensitive than the antigen test were 0.88 and 0.71, respectively. The probabilities that the commercial test was more specific than the saliva and antigen tests were 0.86 and 0.92, respectively (Table S3). The saliva test was more specific than the antigen test with a probability of 0.81.

Joint inference of the test properties and the prevalence of infection allowed us to estimate the frequency of different outcomes from surveillance testing (Fig. 2). Due to its high sensitivity, the saliva test was predicted to yield the most true positives (16.1 per 1,000 tests; 95% CrI, 8.2 to 27.6) (Fig. 2A) and the fewest false negatives (1.3 per 1,000 tests; 95% CrI, 0.07 to 5.4) (Fig. 3C). At the other extreme, 1,000 antigen tests were predicted to yield 12.7 true positives (95% CrI, 5.4 to 23.9) and 4.4 false negatives (95% CrI, 0.5 to 13.1). The antigen tests also had the lowest specificity, resulting in the largest number of false positives (21.3 per 1,000 tests; 95% CrI, 0.8 to 110.3) (Fig. 2B). The commercial tests were estimated to perform best in this regard, yielding only 2.1 false positives per 1,000 tests (95% CrI, 0.1 to 7.5).

Because of their high specificity and the low prevalence of infection, the commercial tests were predicted to have the highest positive predictive value during the study period (87%; 95% CrI, 62 to 99%) (Fig. 3A). Given such low prevalence, all tests had high negative predictive values and were predicted to result in mostly true negatives (Fig. 3B) for the vast majority of tests. Under a scenario of surveillance screening at

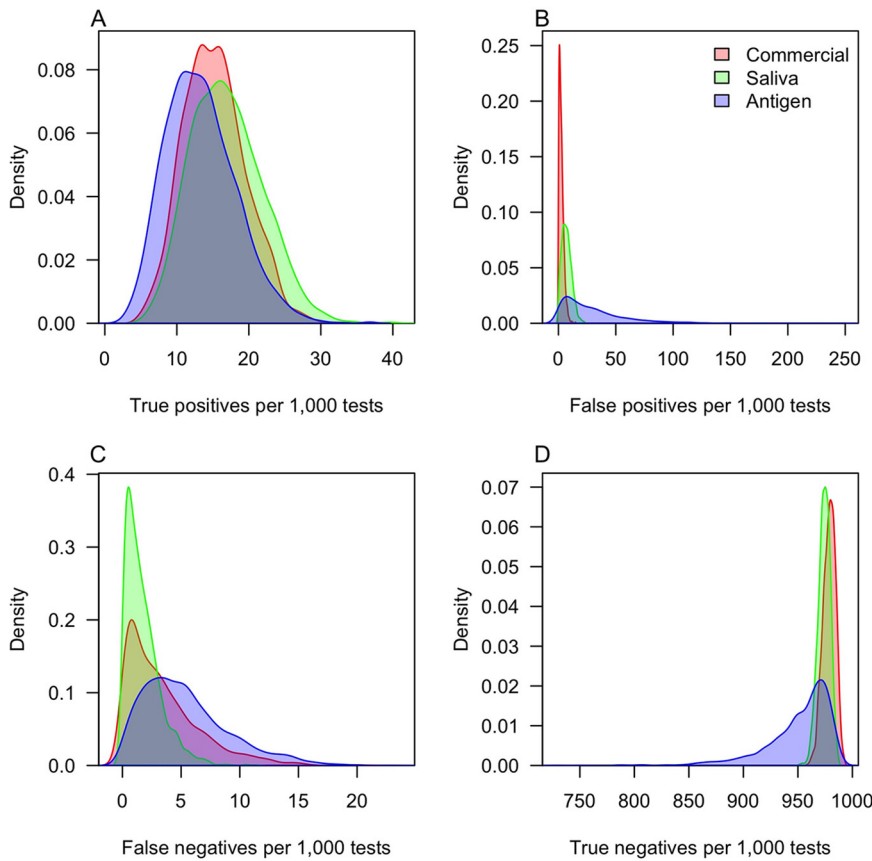

**FIG 2** Estimates of the frequency of different testing outcomes. Out of 1,000 tests, the panels show the number of (A) true positives, (B) false positives, (C) false negatives, and (D) true negatives. The colors distinguish the different types of tests. Values outside the range 0 to 1,000 occur only as a result of smoothing.

random from the campus population during the fall 2020 semester, we estimated that saliva tests during the semester would be expected to have median positive predictive values as low as 0.2% (95% CrI, 0.07 to 2.4%) on 1 August and as high as 89% (95% CrI, 76 to 99%) on 22 August (Fig. 4B). The negative predictive values of the saliva test never would have been less than a median of 99.5% (95% CrI, 98.4 to 99.9%) under this scenario (Fig. 4E). The commercial tests would have had higher positive predictive values under this scenario (Fig. 4A), and both the commercial and antigen tests would have had lower negative predictive values (Fig. 4D and F).

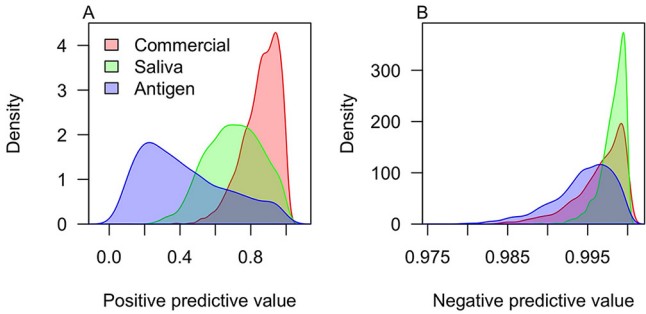

**FIG 3** Estimates of the predictive values of each test during the study period. The panels show estimates of (A) the positive predictive value and (B) the negative predictive value. The colors distinguish the different types of tests. Values outside the range 0 to 1 occur only as a result of smoothing. Decimal values are shown along the *x* axes, consistent with the definitions of these quantities as probabilities, rather than percentages, in Materials and Methods.

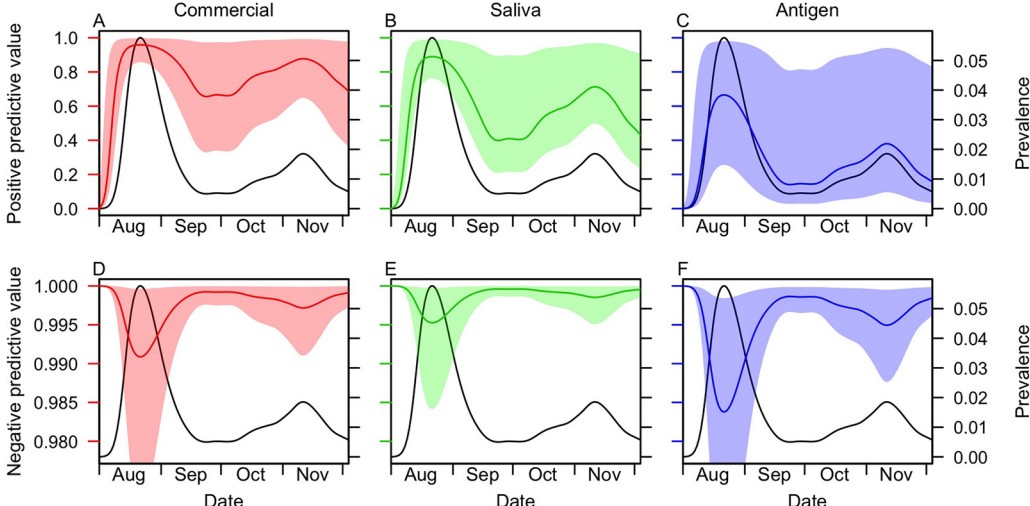

**FIG 4** Positive predictive value (A to C) and negative predictive value (D to F) over the course of the entire semester. These values represent the probability that a positive or negative test result under random surveillance screening would have accurately relayed the true positive or negative status of the individual being tested. The change over time was a result of the time-varying prevalence of detectable infection (black lines, right axis). The uncertainty reflects uncertainty about the sensitivity and specificity of each type of test: commercial (red), saliva (green), and antigen (blue). Decimal values are shown along the *y* axes, consistent with the definitions of these quantities as probabilities, rather than percentages, in Materials and Methods.

## DISCUSSION

With respect to the sensitivity, there were important differences between our modeled estimates and those based on the raw test data. The clearest difference was for the commercial test, for which we obtained a median estimate of 86% for the sensitivity. Had we considered that test to be a gold standard, we would have obtained a point estimate for the sensitivity of the saliva test 16% lower than our median estimate, and 33% lower for the antigen test. On the contrary, we found support for the saliva test likely being more sensitive than the commercial test, which would not have been possible to infer had we assumed a gold standard (20). Our finding that the saliva test was more sensitive than the commercial test, which was based on a nasal swab sample, is consistent with findings from other studies (27–30). We estimated that the antigen test had markedly lower sensitivity than either of the RT-PCR tests, which is also consistent with other studies (31, 32). Given that only 37 individuals received an antigen test, it is important to bear in mind that the uncertainty about its properties is high.

With respect to specificity, the medians of our modeled estimates were slightly lower than estimates based on the commercial test as a gold standard. Even so, our specificity estimate for the saliva test (median, 99.3%; 95% CrI, 98.3 to 99.9%) was strikingly similar to an independent estimate (median, 99.2%; 95% CrI, 98.2 to 99.8%) that also used a BLCM, but in the context of a meta-analysis (23). Our analysis generated high confidence in the commercial test being the most specific and the antigen test being the least specific, with a probability of 0.8. Our median estimate of 97.8% for the antigen test specificity was within the range of published estimates for seven different antigen tests (33), which had median values ranging from 98.5 to 100% for five tests and from 88.9 to 94.8% for two outliers. Uncertainty about the test specificity was relatively low in our estimates, given that the vast majority of the individuals we tested were likely true negatives.

These test sensitivities and specificities have implications for several metrics of public health importance. Given its high sensitivity, the saliva test was expected to detect the most true positives and produce the fewest false negatives, as indicated by a high negative predictive value. For the purpose of identifying infections in surveillance

screening so that they can be isolated and their transmission curtailed, this test was most ideal, especially at times of high prevalence. Given its high specificity, the commercial test was expected to result in the fewest false positives and the most true negatives, as indicated by a high positive predictive value. These properties are ideal from the perspective of minimizing unnecessary demand on resources for case isolation and contact tracing, more so at times of low prevalence when demand is already low. The antigen test performed least well in all regards. While this may make it seem like a less desirable option, it should be noted that sensitivity over the course of infection as a whole need not be paramount. In the event that a test has a high sensitivity around the time of peak infectiousness, its value for curtailing transmission could still be very high (34).

Given the implications of our estimates of sensitivity and specificity, it is important to understand their empirical basis. It is notable that, out of 853 individuals tested, only six had discrepant results. In four of those cases, the saliva test was positive and the commercial test was negative. In two, the commercial test was positive and one or both of the other tests were negative. A strength of our modeling approach is that it integrated across all of the available information to inform its estimates, rather than those six discrepancies alone. The model also took into account the higher positivity of the saliva and antigen tests, compared to the commercial test. Likewise, it was capable of balancing that with indications that the antigen test had a lower specificity, which could explain its higher observed test positivity in part. Additionally, the model was able to account for higher positivity among the nonsurveillance tests due to a higher prevalence in that group, which is important given that the three types of tests were not applied evenly across the two groups. These competing influences on our estimates underscore the value of the BLCM approach we used, which was able to balance them appropriately and express that balance in the form of quantitative descriptions of uncertainty.

Although our analysis was able to provide insight into the properties of these three tests, there were some uncertainties that we were unable to resolve. Correlations among three parameters—$Se_{Comm}$, $Sp_{Saliva}$, and $Prev_{Surv}$—were indicative of uncertainty about whether the four individuals with positive saliva tests and negative commercial tests resulted from false-positive saliva tests or false-negative commercial tests. More data would be helpful for resolving this uncertainty, although doing so would require relatively rare discrepant results. Another limitation of our study is that the only information we used to resolve the uncertainty about the true infection status was whether individuals were tested for surveillance purposes. Given that the prevalence differed by an order of magnitude between these groups, this was quite beneficial. However, additional information—such as recent contacts or status as student, faculty, or staff—could have potentially helped narrow this type of uncertainty further. Doing so would have required estimating more parameters and could have made the analysis more susceptible to bias if an increasingly complex model were not specified properly. A related limitation is that we were unable to examine the possibility of differing sensitivities as a function of the symptom status. Had we incorporated this into our model, it would have been impossible to distinguish between differing sensitivities and differing prevalences among the individuals with differing symptom statuses. Similar to our study, a large-scale study (35) of over 1 million people in the United Kingdom found that the prevalence among individuals presenting with symptoms was several-fold higher than among those not reporting symptoms. Accordingly, we felt that it was appropriate to place more emphasis on estimating the differences in prevalence than the differences in sensitivity between these groups.

In addition to limitations of our analysis, there were also limitations of our data set. First, a more balanced testing effort across different test types and groups of subjects could have helped reduce the uncertainty about certain parameters, especially those relating to the antigen tests. Second, for individuals who truly were positive, we have no information about how many days elapsed between their initial exposure and

when they were tested. Given how variable the test sensitivity is over the course of an infection (11, 17), this factor alone could be a major driver of the sensitivities we estimated. For example, individuals tested for surveillance screening had presumably not displayed any symptoms up to the time of testing, making it possible that many of our positive surveillance tests came from presymptomatic individuals with high viral loads (36). Even so, the balance of saliva and commercial tests across the individuals tested for surveillance versus nonsurveillance purposes was similar, meaning that any differences in the timing of testing between these two groups should not have affected our inferences about the relative sensitivities of these two types of tests.

In conclusion, our analysis leveraged all the data collected from this study to estimate the sensitivities and specificities of three types of tests, without the need to consider any of those tests as a gold standard. These estimates are pertinent to a setting for which surveillance testing has been (37, 38), and remains (39), a major emphasis of COVID-19 prevention. Although there is appreciable uncertainty associated with our estimates, this uncertainty was quantified carefully and could be reduced in the future by updating our estimates with additional data. Bayesian analyses lend themselves to this naturally, given that posterior estimates from one study can serve as prior estimates for another.

## MATERIALS AND METHODS

**Sample collection.** All samples for this study were collected during a 5-day period from Monday, 12 October, through Friday, 16 October 2020. In total, 1,716 tests were performed on samples collected from 853 individuals, with multiple tests for a single individual applied to separate specimens collected on the same day. Most individuals (811) participated in response to a request for surveillance testing, while others (37) participated either as a result of reporting symptoms associated with COVID-19 (40), because of suspected exposure through contact (10), or because they had previously tested positive and were undergoing a second test 4 days later (3). The participants consisted of 846 students and 7 faculty and staff. The majority (87.6%) of the students were between the ages of 18 and 22 inclusive, with a range of 18 to 39, a median age of 20, and a mean age of 21.2. The median age of the staff was 40, and two members of the staff were over the age of 65, with a range of 29 to 72 (Fig. S1). These individuals received a total of 833 commercial RT-PCR tests on nasal swab specimens, 846 in-house RT-PCR tests on saliva specimens, and 37 antigen tests on nasal swab specimens. We refer to these tests here as *commercial*, *saliva*, and *antigen tests*, respectively. A majority of individuals (799) received commercial and saliva tests but not an antigen test, a subset (41) received all three tests, and the remainder (41) received either one or two tests in other combinations. The low number of antigen tests limited the precision of estimates of its performance.

**Laboratory testing. (i) SARS-CoV-2 detection in saliva samples.** Following the University of Notre Dame Institutional Biosafety Committee (IBC)-approved protocol (20-08-6161), fresh saliva samples were obtained from study participants and tested for the presence of SARS-CoV-2 within 17 h after collection. Steps for the detection of SARS-CoV-2 through RT-qPCR in the saliva samples were adapted from Ranoa et al. (41). A minimum of 200 $\mu$L saliva was collected from each participant in a barcoded nuclease-free 50-mL conical tube. Following collection, the samples were heat inactivated by incubating them in a 95°C circulating water bath for 30 min. After cooling to room temperature, the inactive specimen was diluted at a 1:1 ratio (vol/vol) with 2× Tris-borate-EDTA buffer (0.089 M Tris, 0.089 M borate, 0.002 M EDTA in a final 1× buffer solution), followed by vigorous vortexing to ensure thorough mixing. The diluted saliva was then subjected to RT-qPCR (1 reaction per sample) using the TaqPath COVID-19 combo kit (Thermo Fisher Scientific), which includes three primer/probe sets specific to SARS-CoV-2 genes (ORF1ab, S gene, N gene) and one MS2 bacteriophage control target. Briefly, 5 $\mu$L diluted saliva was added to a freshly prepared reaction mix containing 2.5 $\mu$L TaqPath 1-step multiplex master mix (no ROX) (4×), 0.5 $\mu$L COVID-19 real-time PCR assay multiplex, 0.5 $\mu$L MS2 phage control, and 1.5 $\mu$L nuclease-free water. Reactions were set up in a 96-well plate format (0.1 mL MicroAmp fast optical 96-well reaction plate [Applied Biosystems]), with each plate containing a positive control diluted to 4 copies/$\mu$L and a no-template control (nuclease-free water), as well as the MS2 control internal to each sample. All RT-qPCRs were carried out using QuantStudio RT-qPCR instruments (Applied Biosystems). The reaction parameters were set as follows: *hold stage*, 25°C for 2 min, 53°C for 10 min, 95°C for 2 min; *PCR stage* (40×), 95°C for 3 s, 60°C for 30 s; 1.6°C/s ramp for all stages; run mode "fast." Targets, reporter dyes, and quencher information for the RT-qPCR instrument were set up according to the TaqPath COVID-19 combo kit manufacturer's instructions.

A presence/absence analysis of the viral targets was performed using Applied Biosystems Design and Analysis v2.4 software with the baseline set at 5 and the quantification cycle (*Cq*) cutoff for all targets set at 37. The results of the RT-qPCR test were interpreted as positive, negative, or invalid. A positive test had at least 2 of the 3 gene targets present within the threshold settings. All positive and invalid tests were subjected to repeat testing for confirmation.

**(ii) SARS-CoV-2 commercial and rapid antigen detection.** Self-administered nasal swab samples were outsourced to LabCorp, Inc., for viral detection using a RT-PCR protocol (EUA200011). Rapid antigen assays were performed on self-administered nasal swab samples using the Sofia2 fluorescent

immunoassay (FIA) analyzer and the SARS antigen fluorescent immunoassay for qualitative detection of the nucleocapsid protein from SARS-CoV-2 (Quidel).

**Statistical analysis. (i) Model.** For our analysis, we estimated eight parameters (Table 1), which together determine the probability of each type of testing outcome. The likelihood of a given set of values of these parameters is equal to the probability of the observed testing outcomes, given those parameter values. The data were defined according to the number of individuals with a given combination of testing outcomes as $n_{i,j,k}$, where $i$, $j$, and $k$ refer to positive, negative, or missing results, respectively, for each type of test (commercial, saliva, and antigen) (Table 2).

By definition, the test sensitivity and specificity are specified in reference to the true infection status of an individual, which we define as having been infected with SARS-CoV-2 recently enough to still contain sufficient RNA to be detectable. We note that being infected under this definition does not necessarily imply that a person is infectious (42). Because we did not know the true infection status of any individual with certainty, we defined the probability of a given set of testing outcomes (i.e., $i$, $j$, $k$) conditional on the true status, which we refer to as $s$ (this could be either $+$ or $-$). This probability, $Pr(i,j,k|s)$, is defined as the product of the probabilities of each testing outcome given status $s$. It is important to note that our model does not assume or infer the true infection status of any study participants; rather, it treats the true infection status probabilistically as an unknown state.

To account for the fact that the true status of any given infection is unknown, we used the law of total probability to calculate the overall probability of the observed testing outcomes, $Pr(i,j,k)$, as the weighted average of the conditional probabilities of the observed testing outcomes:

$$\Pr(i,j,k) \;=\; \Pr(i,j,k|+)Prev + \Pr(i,j,k|-)\,(1-Prev)$$

Denoting the set of all parameters as $\theta$, we defined the likelihood of the parameters, given the data, **n**, as

$$L(\theta|n) = \prod_{\{i,j,k\}\,\in\,n} Pr(i,j,k)^{n_{i,j,k}}.$$

In these calculations, different values of $Prev$ were used depending on whether the individuals were tested as part of surveillance efforts or for other reasons.

**(ii) Estimation procedure.** Taking a Bayesian approach to parameter estimation, the posterior probability of the parameters in $\theta$ was defined as

$$Pr(\theta|n) \;=\; \frac{L(\theta|n)\,Pr(\theta)}{Pr(n)}$$

where $L(\theta|\mathbf{n})$ is the likelihood defined above, $Pr(\theta)$ is the prior probability of the parameters, and $Pr(\mathbf{n})$ is the probability of the data. We assumed noninformative priors for the sensitivity and specificity parameters, and we assumed informative priors for the two types of prevalence that were in loose alignment with the estimated prevalence at the time and location of sample collection. We avoided calculation of $Pr(\mathbf{n})$ by using Markov chain Monte Carlo (MCMC) sampling. Details about the prior assumptions and MCMC algorithm are provided in the supplemental text.

**(iii) Validation.** To validate our model, we applied it to 100 simulated data sets and compared inferred parameter values to the true parameter values used to simulate the data. To ensure that the model's inferences were valid for data resembling those used in this study, we simulated the same number of individuals tested with the same combination of tests as in our empirical data. Simulated parameter values were drawn uniformly and independently from the 95% credible interval of each parameter. We examined coverage probabilities and correlations between median and true parameter values across the 100 simulated data sets.

**(iv) Predictive value.** Using the posterior parameter estimates, we calculated predictive values of the three tests under two different contexts. These values represent the probability that the test's indication, whether positive or negative, reflects the true status of the individual being tested. The positive predictive value is defined as

$$PPV = Se\,Prev/\big(Se\,Prev + (1-Sp)\,(1-Prev)\big)$$

and the negative predictive value is defined as

$$NPV = Sp(1-Prev)/\big((1-Se)Prev + Sp(1-Prev)\big)$$

First, we calculated PPV and NPV during the 1-week period of our study, accounting for the uncertainty in $Prev$ in doing so. Second, we calculated PPV and NPV on a daily basis over the course of the entire fall 2020 semester, accounting for the daily changes in prevalence over time. Our estimates for time-varying prevalence were based on an extrapolation of the daily incidence of symptomatic cases (40) that accounted for the proportion symptomatic (43), the incubation period (44), and the probability that a test administered on a given day of infection would be positive (11). This method is described in further detail in the supplemental text.

**Data availability.** All code and data from this study are available at https://github.com/TAlexPerkins/SARSCoV2_BLCM to facilitate applications or extensions of this work by others.

## SUPPLEMENTAL MATERIAL

Supplemental material is available online only.

**SUPPLEMENTAL FILE 1**, PDF file, 0.7 MB.

## ACKNOWLEDGMENTS

We thank Paul Hergenrother for generously sharing his expertise and experience designing and implementing a saliva-based COVID-19 surveillance program at Illinois University, Champaign-Urbana. Special thanks to Joanna McNulty and Carol Mullaney for coordinating the sample collection efforts and the technicians in the Notre Dame COVID Surveillance Lab who assisted with sample processing and data collection. This study was made possible by the enthusiastic participation of Notre Dame undergraduate students, staff, and faculty.

Funding was provided by the University of Notre Dame and Notre Dame Research. This study was conducted under University of Notre Dame IRB number 21-06-6687.

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
