## [Reviewer comments · Microbiology Spectrum]

Microbiology Spectrum

Performance of three tests for SARS-CoV-2 on a university campus estimated jointly with Bayesian latent class modeling

Alex Perkins, Melissa Stephens, Wendy Alvarez Barrios, Sean Cavany, Liz Rulli, and Michael Pfrender

Corresponding Author(s): Alex Perkins, University of Notre Dame

Review Timeline:

Submission Date:	August 9, 2021
Editorial Decision:	September 7, 2021
Revision Received:	November 29, 2021
Accepted:	December 12, 2021

Editor: Jennifer Dien Bard

Reviewer(s): The reviewers have opted to remain anonymous.

Transaction Report:

DOI: <https://doi.org/10.1128/spectrum.01220-21>

September 7, 2021

Dr. Alex Perkins
University of Notre Dame
Notre Dame, IN

Re: Spectrum01220-21 (Performance of three molecular tests for SARS-CoV-2 on a university campus estimated jointly with Bayesian latent class modeling)

Dear Dr. Alex Perkins:

Thank you for submitting your manuscript to Microbiology Spectrum. The manuscript was reviewed by two experts in the field and it was deemed that it may be appropriate for publication if the comments provided are appropriately addressed. Please pay close attention to the comments provided by Reviewer #1. When submitting the revised version of your paper, please provide (1) point-by-point responses to the issues raised by the reviewers as file type "Response to Reviewers," not in your cover letter, and (2) a PDF file that indicates the changes from the original submission (by highlighting or underlining the changes) as file type "Marked Up Manuscript - For Review Only". Please use this link to submit your revised manuscript - we strongly recommend that you submit your paper within the next 60 days or reach out to me. Detailed information on submitting your revised paper are below.

Link Not Available

Sincerely,

Jennifer Dien Bard

Journals Department
Reviewer comments:

Reviewer #1 (Comments for the Author):

Perkins et al. used an alternative approach such as Bayesian latent class modeling to estimate the accuracy of different tests for SARS-CoV-2. They applied this technique to a collection of >1700 patient samples consisting of different specimen types and were tested using different platforms. Their modeling data show that testing of SARS-CoV-2 using RT-PCR on saliva specimens is an effective testing surveillance strategy.

Major comments:

1. Lines 51-55: This manuscript here challenges the need for a gold standard and explains how using a modeling approach may circumvent that. In lines 51-53, the sentence reads as if methods of sample extraction, day of infection, and disease severity are reasons why there isn't a good gold standard test for SARS-CoV-2. However, using your modeling system which extrapolates data from these current testing platforms that are affected by such factors does not fully support the notion that your modeling is not influenced by such limitations either.
2. Although the performance of antigen testing has been shown to be poorer compared to RT-PCR by many other studies, it is concerning that the sweeping claims about the accuracy of antigen testing here are done on 37 tests only compared to the 846 RT-PCR tests. The low power in the sample size is an issue.
3. It should be noted if the specimens tested on the specific platforms were validated specimens as that would affect the

performance characteristics (e.g. did you use saliva on the antigen test and if so, was saliva validated?)

4. Line 74-75: May be an overstatement to say that samples from 18-25 yo are less sensitive. More supporting citations?
5. Line 158: Calculations for test sensitivity and test specificity are in reference to true infection status but it is dangerous to make such a claim and assumption in the modeling parameters. The concerning part of this pandemic is that asymptomatic individuals can also spread the infection, those who test positive may have lingering nucleic acid rendering them not infectious, and there is actually no test for infectiousness altogether. How you are defining as "true infection status" will change your results. Meanwhile, in reality, when individuals get tested, infection status is commonly not a priori knowledge.
6. Calculations for PPV and NPV on a daily basis over the course of the entire fall 2020 semester are done using data extrapolated from symptomatic cases. However, asymptomatic individuals can also spread SARS-CoV-2 and be infected. They, too, should be a part of the calculation.
7. Were there differences in the accuracy of the test between asymptomatic and symptomatic individuals?
8. Line 297-298: Estimates of PPV for August 1 and August 22 are written. Given that we have passed this date, how accurate is the modeling per actual data?
9. Were there differences in the accuracy of the test between asymptomatic and symptomatic individuals?
10. Line 297-298: Estimates of PPV for August 1 and August 22 are written. Given that we have passed this date, how accurate is the modeling per actual data?

Minor comments:

1. Data presentation: The data presented in this manuscript do not follow conventional formatting and may be confusing for the audience. Sensitivity/specificity/PPV/NPV are usually expressed in %s (table 1) and all the figures could be made into tables with the actual %s listed (range can be noted as well if necessary).
2. Line 225: "Pooling"-during this pandemic, the word "pooling" has been used to describe an approach to testing specimens. Please use another word here or if you are using "pooling" (to describe combination of multiple patient samples, as it has been during this pandemic) please clarify the methods/results.
3. A table that combining the information from Table 2 and Figure S1 would be beneficial and will offer more information (e.g. number of specimen types associated with age group).

Reviewer #2 (Comments for the Author):

The manuscript entitled "Performance of three molecular tests for SARS-CoV-2 on a university campus estimated jointly with Bayesian latent class modeling" by Perkins et al. describes the use of Bayesian latent class modeling (BLCM) to estimate the accuracy of three SARS-CoV-2 diagnostic tests: two nucleic acid amplification tests (NAAT; one nasal [commercial] and the other a saliva test [performed in-house]) and an antigen test. The authors found that the in-house saliva test performed the best, followed by the commercial NAAT, and finally the antigen test. The authors' data suggested that NAAT testing of saliva samples by their in-house laboratory was effective for SARS-CoV-2 surveillance screening.

Overall, I feel that the work described in the manuscript was relevant and very well done. The data was well-presented and described, and the entire manuscript was very well written.

The only modification that I would like to see made to this manuscript is changing the title: the current title states that three molecular tests were compared; in reality, two molecular tests and one antigen test were compared. This seems like a trivial matter; however, in the diagnostic and public health laboratory microbiology communities, antigens tests are NOT considered to be molecular tests. Rather, the designation of "molecular" test is only used for assays that detect nucleic acids (e.g., NAATs).

Staff Comments:

Preparing Revision Guidelines

Please return the manuscript within 60 days; if you cannot complete the modification within this time period, please contact me. If you do not wish to modify the manuscript and prefer to submit it to another journal, please notify me of your decision immediately so that the manuscript may be formally withdrawn from consideration by Microbiology Spectrum.

Perkins et al. used an alternative approach such as Bayesian latent class modeling to estimate the accuracy of different tests for SARS-CoV-2. They applied this technique to a collection of >1700 patient samples consisting of different specimen types and were tested using different platforms. Their modeling data show that testing of SARS-CoV-2 using RT-PCR on saliva specimens is an effective testing surveillance strategy.

Major comments:

1. Lines 51-55: This manuscript here challenges the need for a gold standard and explains how using a modeling approach may circumvent that. In lines 51-53, the sentence reads as if methods of sample extraction, day of infection, and disease severity are reasons why there isn't a good gold standard test for SARS-CoV-2. However, using your modeling system which extrapolates data from these current testing platforms that are affected by such factors does not fully support the notion that your modeling is not influenced by such limitations either.
2. Although the performance of antigen testing has been shown to be poorer compared to RT-PCR by many other studies, it is concerning that the sweeping claims about the accuracy of antigen testing here are done on 37 tests only compared to the 846 RT-PCR tests. The low power in the sample size is an issue.
3. It should be noted if the specimens tested on the specific platforms were validated specimens as that would affect the performance characteristics (e.g. did you use saliva on the antigen test and if so, was saliva validated?)
4. Line 74-75: May be an overstatement to say that samples from 18-25 yo are less sensitive. More supporting citations?
5. Line 158: Calculations for test sensitivity and test specificity are in reference to true infection status but it is dangerous to make such a claim and assumption in the modeling parameters. The concerning part of this pandemic is that asymptomatic individuals can also spread the infection, those who test positive may have lingering nucleic acid rendering them not infectious, and there is actually no test for infectiousness altogether. How you are defining as "true infection status" will change your results. Meanwhile, in reality, when individuals get tested, infection status is commonly not a priori knowledge.
6. Calculations for PPV and NPV on a daily basis over the course of the entire fall 2020 semester are done using data extrapolated from symptomatic cases. However, asymptomatic individuals can also spread SARS-CoV-2 and be infected. They, too, should be a part of the calculation.
7. Were there differences in the accuracy of the test between asymptomatic and symptomatic individuals?
8. Line 297-298: Estimates of PPV for August 1 and August 22 are written. Given that we have passed this date, how accurate is the modeling per actual data?
9. Were there differences in the accuracy of the test between asymptomatic and symptomatic individuals?
10. Line 297-298: Estimates of PPV for August 1 and August 22 are written. Given that we have passed this date, how accurate is the modeling per actual data?

Minor comments:

1. Data presentation: The data presented in this manuscript do not follow conventional formatting and may be confusing for the audience. Sensitivity/specificity/PPV/NPV are usually expressed in

%s (table 1) and all the figures could be made into tables with the actual %s listed (range can be noted as well if necessary).

2. Line 225: “Pooling”—during this pandemic, the word “pooling” has been used to describe an approach to testing specimens. Please use another word here or if you are using “pooling” (to describe combination of multiple patient samples, as it has been during this pandemic) please clarify the methods/results.
3. A table that combining the information from Table 2 and Figure S1 would be beneficial and will offer more information (e.g. number of specimen types associated with age group).

Reviewer #1 (Comments for the Author):

Perkins et al. used an alternative approach such as Bayesian latent class modeling to estimate the accuracy of different tests for SARS-CoV-2. They applied this technique to a collection of >1700 patient samples consisting of different specimen types and were tested using different platforms. Their modeling data show that testing of SARS-CoV-2 using RT-PCR on saliva specimens is an effective testing surveillance strategy.

Major comments:

1. Lines 51-55: This manuscript here challenges the need for a gold standard and explains how using a modeling approach may circumvent that. In lines 51-53, the sentence reads as if methods of sample extraction, day of infection, and disease severity are reasons why there isn't a good gold standard test for SARS-CoV-2. However, using your modeling system which extrapolates data from these current testing platforms that are affected by such factors does not fully support the notion that your modeling is not influenced by such limitations either.

Response: The reviewer raises a fair point that these issues reduce the reliability of test results and that our modeling approach does not ameliorate those issues directly. However, our modeling approach does offer an improvement for dealing with those issues, as it acknowledges—and explicitly quantifies—the inaccuracies in test results associated with them. To make our view on this issue clearer, we have added the following sentence to the paragraph that follows the passage that the reviewer's comment pertains to.

“While this approach does not make test results more accurate per se, it does reduce the risk of bias associated with erroneously assuming that a gold standard is without error.”

2. Although the performance of antigen testing has been shown to be poorer compared to RT-PCR by many other studies, it is concerning that the sweeping claims about the accuracy of antigen testing here are done on 37 tests only compared to the 846 RT-PCR tests. The low power in the sample size is an issue.

Response: We agree with the reviewer that it is important to convey the limitations of our inferences about the performance of the antigen test given the small sample size. In addition to places in the manuscript where this was already mentioned, we have drawn additional attention to this issue with the following additions.

Abstract: *“An antigen test was less sensitive and specific than both of the RT-PCR tests, although sample sizes with this test were low and statistical uncertainty was high.”*

Methods: *“The low number of antigen tests limited the precision of estimates of its performance.”*

Results: ***“It is important to note the large uncertainty around these estimates due to the relatively low number of individuals who received an antigen test.”***

3. It should be noted if the specimens tested on the specific platforms were validated specimens as that would affect the performance characteristics (e.g. did you use saliva on the antigen test and if so, was saliva validated?)

Response: For all test comparisons, multiple samples were collected from the same individual at the same time. However, the sample types and the tests are different. Commercial RT-PCR and antigen tests were conducted on samples collected through self-administered nasal swabs, not saliva. We are limited in our ability to isolate the effects of sample type from test platform in our comparison. To clarify this point in the text, we made a slight modification to a sentence from the Methods section that now reads as follows.

*“multiple tests for a single individual applied to **separate** specimens collected on the same day”*

4. Line 74-75: May be an overstatement to say that samples from 18-25 yo are less sensitive. More supporting citations?

Response: We appreciate the reviewer’s perspective on this and have softened the wording from saying “may be less sensitive” to “could potentially be less sensitive.” We have cited two additional references. When taken together, these references are consistent with our wording that implies that understanding of this issue remains somewhat unclear.

5. Line 158: Calculations for test sensitivity and test specificity are in reference to true infection status but it is dangerous to make such a claim and assumption in the modeling parameters. The concerning part of this pandemic is that asymptomatic individuals can also spread the infection, those who test positive may have lingering nucleic acid rendering them not infectious, and there is actually no test for infectiousness altogether. How you are defining as “true infection status” will change your results. Meanwhile, in reality, when individuals get tested, infection status is commonly not a priori knowledge.

Response: We have added the text below near this passage to clarify our view about the model and its connection to an individual’s true infection status.

“It is important to note that our model does not assume or infer the true infection status of any study participants; rather, it treats true infection status probabilistically as an unknown state.”

In general, we view the inference of an individual’s true infection status with humility, and we believe that acknowledging the inherent uncertainty about this is a far less dangerous way to approach this issue than to regard the outcome of a given test as being definitive.

Regarding the issue of infectiousness, we agree with the reviewer and do not intend to make any claims or definitive linkages between infection status and infectiousness. To clarify this, we have revised this line in the manuscript as follows.

“By definition, test sensitivity and specificity are specified in reference to the true infection status of an individual, which we define as having been infected with SARS-CoV-2 recently enough to still contain sufficient RNA to be detectable. We note that being infected under this definition does not necessarily imply that a person is infectious (28).”

6. Calculations for PPV and NPV on a daily basis over the course of the entire fall 2020 semester are done using data extrapolated from symptomatic cases. However, asymptomatic individuals can also spread SARS-CoV-2 and be infected. They, too, should be a part of the calculation.

Response: We agree with the reviewer that asymptomatic individuals should be accounted for in our estimates of time-varying prevalence of infection, to the extent that doing so is possible. As described in the Supplemental Text (subsection titled “Estimation of time-varying prevalence”), we accounted for asymptomatic infections through an extrapolation that assumes that only 57% of all infections present with symptoms. The figure of 57% was taken from a high-quality study of an outbreak of a demographically similar population of sailors on a U.S. Navy ship that experienced an outbreak. While it would have been more ideal to have data directly on asymptomatic infections on Notre Dame’s campus, the use of surveillance testing was inconsistent over the course of the semester and, therefore, cannot be used as a reliable measure of temporal patterns of asymptomatic infections over the course of the semester. One additional piece of information that we find reassuring, however, is that the temporal pattern of symptomatic cases closely resembles temporal patterns in SARS-CoV-2 RNA concentration in wastewater samples collected throughout the semester. In the revised manuscript, we drew attention to this through the following addition to the Supplemental Text.

“Although it would have been more ideal to make use of data that speaks directly to asymptomatic infections, the use of surveillance testing was too inconsistent over the course of the semester to inform estimates of time-varying patterns of asymptomatic infection incidence. However, one independent data stream that supports our extrapolation of time-varying patterns of symptomatic infections comes from SARS-CoV-2 RNA concentrations from wastewater samples, which were collected consistently throughout the semester and display similar trends as symptomatic infection incidence (29).”

Last, we note that while the description of this aspect of our methods was correct in the previous version of the manuscript, we had mistakenly failed to account for the extrapolation involving asymptomatic infections in our calculations. We have since rectified this and updated the results presented in Figure 4 accordingly, as well as the text that refers to it. In brief, this adjustment slightly increased the positive predictive values and slightly decreased the negative predictive values, due to the slightly higher prevalence assumed.

7. Were there differences in the accuracy of the test between asymptomatic and symptomatic individuals?

Response: This is an interesting question, and we suspect that there may be some interesting differences between the sensitivity of asymptomatic and symptomatic individuals. However, we did not have confidence in our ability to estimate such a difference, for reasons explained in the following excerpt, which we have added to the Discussion in response to this comment.

“A related limitation is that we were unable to examine the possibility of differing sensitivities as a function of symptom status. Had we incorporated this into our model, it would have been impossible to distinguish between differing sensitivities and differing prevalences among individuals with differing symptom statuses. Similar to our study, a large-scale study (40) of over 1 million people in the United Kingdom found that prevalence among individuals presenting with symptoms was several fold higher than among those not reporting symptoms. Accordingly, we felt that it was appropriate to place more emphasis on estimating differences in prevalence than differences in sensitivity between these groups.”

8. Line 297-298: Estimates of PPV for August 1 and August 22 are written. Given that we have passed this date, how accurate is the modeling per actual data?

Response: We do not have data that would allow us to evaluate this. The only time during the semester when data involving multiple tests on the same samples were available was during the one-week period in October that our primary analysis is based on. In contrast, the time-varying estimates of predictive value were based on estimates of sensitivity and specificity from our primary analysis and estimates of time-varying prevalence extrapolated from time-varying incidence of symptomatic infections.

Minor comments:

1. Data presentation: The data presented in this manuscript do not follow conventional formatting and may be confusing for the audience. Sensitivity/specificity/PPV/NPV are usually expressed in %s (table 1) and all the figures could be made into tables with the actual %s listed (range can be noted as well if necessary).

Response: Thank you for noting this. In response, we changed the presentation of these quantities in the text such that they now read as percentages. We hope that this may feel more familiar to a broader set of readers. At the same time, we have retained our use of decimal values of these quantities in Table 1 and Figures 1, 3, and 4. We added a sentence to the caption of each of those to describe our justification for doing so, with an example of one of those sentences below.

“Decimal values are shown along the x-axes, consistent with the definitions of these quantities as probabilities, rather than percentages, in the Methods section.”

2. Line 225: "Pooling"-during this pandemic, the word "pooling" has been used to describe an approach to testing specimens. Please use another word here or if you are using "pooling" (to describe combination of multiple patient samples, as it has been during this pandemic) please clarify the methods/results.

Response: We reworded this to "Combining data."

3. A table that combining the information from Table 2 and Figure S1 would be beneficial and will offer more information (e.g. number of specimen types associated with age group).

Response: Although we agree with the reviewer that it could be interesting to explore the possibility of differential test performance between different age groups, we do not have ready access to the data that would be needed to explore that. We sought permission from our institutional review board to obtain data on study participant age only in aggregate rather than linking it to data about specific tests. Given that we did not explore differences in test performance by study participant age, presentation of data stratified in this manner does not seem of primary importance. We assume that the reviewer may feel similarly given that this was listed in the "Minor comments" section of their review. Last, we note that there were only 33 study participants of age 30 or older, meaning that there is almost certainly insufficient data in that older age group to support any differences in our conclusions with respect to study participant age.

Reviewer #2 (Comments for the Author):

The manuscript entitled "Performance of three molecular tests for SARS-CoV-2 on a university campus estimated jointly with Bayesian latent class modeling" by Perkins et al. describes the use of Bayesian latent class modeling (BLCM) to estimate the accuracy of three SARS-CoV-2 diagnostic tests: two nucleic acid amplification tests (NAAT; one nasal [commercial] and the other a saliva test [performed in-house]) and an antigen test. The authors found that the in-house saliva test performed the best, followed by the commercial NAAT, and finally the antigen test. The authors' data suggested that NAAT testing of saliva samples by their in-house laboratory was effective for SARS-CoV-2 surveillance screening.

Overall, I feel that the work described in the manuscript was relevant and very well done. The data was well-presented and described, and the entire manuscript was very well written.

Response: We thank the reviewer for their supportive comments.

The only modification that I would like to see made to this manuscript is changing the title: the current title states that three molecular tests were compared; in reality, two molecular tests and one antigen test were compared. This seems like a trivial matter; however, in the diagnostic and public health laboratory microbiology communities, antigens tests are NOT considered to be

molecular tests. Rather, the designation of "molecular" test is only used for assays that detect nucleic acids (e.g., NAATs).

Response: In response to this comment, we have removed the word "molecular" from the title. We also removed this word from a place in the Abstract where it had been used similarly in the previous version of the manuscript.

November 30, 2021

Dr. Alex Perkins
University of Notre Dame
Notre Dame, IN

Re: Spectrum01220-21R1 (Performance of three tests for SARS-CoV-2 on a university campus estimated jointly with Bayesian latent class modeling)

Dear Dr. Alex Perkins:

I am pleased to share that your manuscript has been accepted, and I am forwarding it to the ASM Journals Department for publication. You will be notified when your proofs are ready to be viewed.

Sincerely,

Jennifer Dien Bard
Editor, Microbiology Spectrum
